# Optimal Greedy Control in Reinforcement Learning

**DOI:** 10.3390/s22228920

**Published:** 2022-11-18

**Authors:** Alexander Gorobtsov, Oleg Sychev, Yulia Orlova, Evgeniy Smirnov, Olga Grigoreva, Alexander Bochkin, Marina Andreeva

**Affiliations:** 1Higher Mathematics Department, Volgograd State Technical University, Lenin Ave, 28, Volgograd 400005, Russia; 2Mechanical Engineering Research Institute, Russian Academy of Sciences, Maly Kharitonyevsky Pereulok, 4, Moscow 101990, Russia; 3Software Engineering Department, Volgograd State Technical University, Lenin Ave, 28, Volgograd 400005, Russia

**Keywords:** optimal control, variational methods, machine learning, reinforcement learning, robotics

## Abstract

We consider the problem of dimensionality reduction of state space in the variational approach to the optimal control problem, in particular, in the reinforcement learning method. The control problem is described by differential algebraic equations consisting of nonlinear differential equations and algebraic constraint equations interconnected with Lagrange multipliers. The proposed method is based on changing the Lagrange multipliers of one subset based on the Lagrange multipliers of another subset. We present examples of the application of the proposed method in robotics and vibration isolation in transport vehicles. The method is implemented in FRUND—a multibody system dynamics software package.

## 1. Introduction and Related Works

The problem of optimal control is an important scientific problem in various fields of technology, e.g., robotics, vibration damping systems, etc. The exact theoretical solution to this problem can be achieved by using Bellman’s dynamic programming method [1] and Pontryagin’s maximum principle [2]. However, these methods are limited to low-dimensional equations because of their high computational complexity. Today, various variational formulations of optimal control problems, in particular, reinforcement learning, have become widely used. When using this approach, the control problem is simplified by parametrizing the control function and reducing it to the parametric optimization problem. However, a number of topical control problems (for example, in robotics) still have too high dimensionality to be solved efficiently [3,4,5,6,7,8,9,10,11]. Some studies (e.g., We et al. [12] and Tu Vu et al. [13]) have investigated the stability problem of perturbed control motion for known referenced motion, which is a much easier task. Our study is aimed at finding the reference motion.

In the general form, the optimal control problem has the following formulation [1]. For the system described by differential equations
(1)$$f(\dot{x},x,u,t)=0,$$
where x(t) is the coordinate vector of the entire system with dimension n. We need to find control functions, u(t), that let us achieve the extreme value of the criterion
(2)$$I=\int_{0}^{T}R\left(\dot{x},x,u,t\right)dt.$$

As a rule, sign-constant functions are used as the *R* function. It was previously noted that the exact solution for Equations (Equation 1) and (Equation 2) using Bellman’s dynamic programming method and Pontryagin’s maximum principle was only obtained for several cases of low-dimension tasks [1,2]. The optimal control problem (Equation 1 and Equation 2) is transformed into the parametric optimization problem in the reinforcement learning method by using a discrete form of recording the optimization criterion and parameterizing the control function. The optimization criterion is [14]: (3)$$I=\sum_{i=0}^{i=N}R_{i}\left(\dot{x},x,u,t\right)\gamma^{i},$$
where $R_{i}$ is the value of the criterion function corresponding to the *i*-th moment of time, and γ is the discount coefficient, which takes a value from 0 to 1. It is assumed that the time interval of control *T* is divided into *N* sections. The control function is parameterized on basic functions and takes the form u(s,t), where s is the parameter of the control function. Neural networks, Fourier series, etc., can be used as basic functions [14]. The discount coefficient γ allows the optimality criterion to “weaken” (Equation 2). Formula (Equation 3) corresponds to a discrete formulation of Bellman’s optimal control problem when the discount coefficient is equal to one. The control function is called “greedy” in the reinforcement learning method when it was obtained with a discount coefficient equal to zero. The parameterization of the control function for multidimensional problems leads to high-dimensional optimization problems and also makes the solution dependent on the basic functions on which the control function was interpolated. Moreover, the dependence of the control function on time ties it to time-dependent external disturbances. All this makes developing new methods of solving variational formulations of machine learning problems important.

## 2. Theoretical Description

Consider the optimal control problem for systems with constraints in the form of algebraic equations. The equations of the state of these systems can be written as
(4)$$\begin{array}{c} f(\dot{x},x,u,t)=0\\ Q(x,t)=0, \end{array}$$
where Q(x,t) is the constraint equation vector with dimensionality k≤n. For the numerical solution, system (Equation 4) is usually used in the form [15,16,17]
(5)$$\begin{array}{c} f\left(\dot{x},x,u,t\right)+D^{T}p=0\\ D\dot{x}=h\left(x,t\right), \end{array}$$
where D is the matrix of coefficients of the constraint equations with dimension k×n, p is the *k*-dimensional vector of Lagrange multipliers, and h(x,t) is the vector of the right parts of derivatives of the constraint equations. The second equation of system (Equation 5) is obtained by differentiating the constraint equations with respect to time. The physical meaning of the Lagrange multipliers for the problems of the dynamics of mechanical systems is the constraint reactions. As the applications considered in this article are related to mechanical systems, the term “constraint reactions” will be used as equal to the term “Lagrange multipliers”.

The differential-algebraic system in Equation (Equation 5) is widely used in multibody systems (MBS) dynamics software packages for modeling the dynamics of connected systems of bodies [16]. The features of numerical integration (Equation 5) related to ensuring stability are considered in [17]. In numerical integration (Equation 5), derivatives of coordinates and Lagrange multipliers from the system of linear algebraic equations are found at each integration step according to
(6)MDTD0x˙p=f*(x,t)h(x,t).

The solution to a system of linear algebraic equations can be written in the following form
(7)x˙p=A−1b.

Consider the control problem as described in [18]. There is a subset of reactions $p_{1}$ in the vector of constraint reactions p whose elements are numbered from the set $K_{1}$; the number of elements in the set $K_{1}$ is $k_{1}$.Their values are described by functions $\varphi_{i}\left(t\right)$, $i=1,2,\ldots ,k_{1}$, or in the matrix form
(8)$$p_{1}=\varphi \left(t\right).$$

There is also a subset of $k_{2}$ reactions, $p_{2}$, from the vector of constraint reactions, p, whose values are taken from subset $K_{2}$, which can vary due to changes in the values of unknown functions $h_{2j}\left(t\right)$, $j=1,2,\ldots ,k_{2}$, which will be called corrective terms. Each reaction from $p_{2}$ corresponds to its own constraint equation; the relevant corrective term $h_{2j}\left(t\right)$ is added to the right part of this equation. The corrective terms $h_{2j}\left(t\right)$ form a column matrix $h_{2}$ in the $k_{2}$ dimension. Reactions $p_{2}$, generally, are control functions, so we will consider $k_{2}$ as the number of control functions. The values of reactions $p_{1}$, taking into account (Equation 7) and the corrective terms, are
(9)$$p_{1}=A_{1}^{-1}\left(b+h_{2}^{*}\left(t\right)\right),$$
where $A_{1}^{-1}$ corresponding to set $K_{1}$ is the submatrix of $A^{-1}$, consisting of the rows $A^{-1}$ whose numbers belong to $K_{1}$. Only the components with numbers from $K_{2}$ are non-zero in the column matrix $h_{2}^{*}$ of dimension n+k. We assume that $p_{10}=A_{1}^{-1}b$, $A_{1}^{-1}h_{2}^{*}=Ch_{2}$, matrix C contains only columns from matrix $A_{1}^{-1}$ with numbers from $K_{2}$ and has dimensionality $k_{1}\times k_{2}$. Then (Equation 9) takes the form
(10)$$p_{1}=p_{10}+{\mathrm{Ch}}_{2}\left(t\right).$$

Taking into account that $p_{1}=\varphi \left(t\right)$, from (Equation 10), we can obtain the system of linear algebraic equations for determining $h_{2}\left(t\right)$.
(11)$${\mathrm{Ch}}_{2}\left(t\right)=\varphi \left(t\right)-p_{10}.$$

If the system of linear Equation (Equation 11) is joint, then it is possible to determine the control functions of $p_{2}$ as
(12)$$p_{2}=A_{2}^{-1}\left({b+h}_{2}^{*}\left(t\right)\right),$$
where $A_{2}^{-1}$ is the submatrix $A^{-1}$ corresponding to set $K_{2}$, consisting of rows $A^{-1}$ whose numbers belong to $K_{2}$. Given that $p_{20}=A_{2}^{-1}b$, $A_{2}^{-1}h_{2}^{*}={\mathrm{Bh}}_{2}$, matrix B contains only columns from matrix $A_{2}^{-1}$ with numbers from $K_{2}$ and has dimension $k_{2}\times k_{2}$. Equation (Equation 12) can be rewritten as
(13)$$p_{2}=p_{20}+{\mathrm{Bh}}_{2}\left(t\right).$$

Equation (Equation 11) gives the values of changes on the right sides of the constraint equations, ensuring the achievement of the desired values of reactions $p_{1}$. Since $h_{2}\left(t\right)$ affects all the variables in the system (Equation 4), when integrating the equations of the mathematical model, the accelerations and constraint reactions are calculated from the system with the modified right side as follows
(14)x˙p=A−1(b+h2*(t)).

Equation (Equation 11) has a unique solution if $k_{1}=k_{2}$ and matrix C is non-singular. The properties of matrix C are determined by the properties of matrix A. The main reason for the singularity of matrix A is redundant constraints, i.e., linearly dependent rows in matrix D. Redundant constraints can be inherent to the system’s structure or introduced on purpose, for example, in the parallel-structure mechanisms. In the following, it is assumed that the square matrix A is non-singular unless otherwise stated.

Case $k_{1}=k_{2}$ is the simplest. Cases $k_{1}\neq k_{2}$ are of greater interest, so we will consider them further.

The systems where $k_{1}>k_{2}$ are commonly called underactuated systems. The analysis of the controlled motion of these systems can be found in [19]. It is difficult to obtain a meaningful solution for such systems within the framework of the considered approach.

Systems where $k_{1}>k_{2}$ are called overactuated. These systems are widespread, for example, in robotics. Their analysis is relevant to our study. We consider the methods of solving the linear system of equations (Equation 11) in this case. Matrix C of the system is rectangular—with dimensions $k_{1}\times k_{2}$. As already mentioned, matrix C is a full-rank matrix.

System (Equation 12) can be converted to a system with a square matrix by adding equations. The simplest way of achieving this is by adding linear equations for the corrective terms $h_{2}$, i.e., converting (Equation 11) to the form
(15)CV1h2(t)=φ(t)−p10b1,
where $V_{1}$ is the non-singular matrix of constant terms with dimensionality $\left(k_{2}-k_{1}\right)\times k_{2}$, and **b**${}_{1}$ is the column matrix of constant terms on the right side. Only $\left(k_{2}-k_{1}\right)\times \left(k_{1}+1\right)$ terms are linearly independent in the second equation of the system (Equation 15), so matrix $V_{1}$ can be represented as
(16)$$V_{1}=\left(\begin{array}{cc} E & V_{1}^{*} \end{array}\right),$$
where E is the identity matrix of dimensionality $\left(k_{2}-k_{1}\right)\times \left(k_{2}-k_{1}\right)$, and $V_{1}^{*}$ is the matrix of arbitrary coefficients of dimensionality $\left(k_{2}-k_{1}\right)\times k_{1}$. This method of reduction to a single solution will be called the method of additional equations for corrective terms.

Additional equations to (Equation 11) can be formed by imposing linear connections on the controls. The second equation of system (Equation 15) will take the form
$$V_{1}p_{2}=b_{1}.$$

Substituting $p_{2}=p_{20}+{\mathrm{Bh}}_{2}\left(t\right)$, we will get
$$V_{1}{\mathrm{Bh}}_{2}\left(t\right)=b_{1}-V_{1}p_{20}.$$

System (Equation 15) is now
(17)CV1h2(t)=φ(t)−p10b1−V1p20.

We call (Equation 17) the method of reduction to a single solution by additional equations for controls.

Another way to eliminate the uncertainty of solution (Equation 11) is the conditional extremum method. Consider the conditions for the extremum of the expression
(18)$$I=p_{2}^{T}V_{2}p_{2},$$
where $V_{2}$ is a diagonal matrix of weights. Therefore, (Equation 18) is the weighted sum of the squares of controls. Consider the problem of finding the conditional extremum of expression (Equation 18), taking into account the conditions (Equation 11). In this case, (Equation 18) will be represented as
(19)$$I^{*}=p_{2}^{T}V_{2}p_{2}+\left({\mathrm{Ch}}_{2}\left(t\right)-\varphi \left(t\right)-p_{10}\right)\lambda ,$$
where λ is the column matrix of Lagrange multipliers of dimension $k_{1}$. Extremum conditions for (Equation 20) are
(20)$$\frac{\partial I^{*}}{\partial h_{2i}}=0,i=1,2,\ldots ,k_{2},$$
from which we get
(21)B1CTC0=h2λ=b21b22,
where $B_{1}$ is the matrix of dimension $k_{2}\times k_{2}$ with elements
(22)$$b_{1lm}=2v_{2mm}\sum_{i=1}^{k_{2}}b_{im}b_{lm},l,m=1,2,\ldots ,k_{2},$$
the column matrix $b_{21}$ of dimension $k_{2}$ with elements is
(23)$$b_{21l}=2\sum_{i=1}^{k_{2}}v_{2ii}p_{20i}b_{li},l=1,2,\ldots ,k_{2},$$
the column matrix $b_{22}$ of dimension $k_{1}$ is
(24)$$b_{22}=\varphi \left(t\right)+p_{10}.$$

The system of linear equations (Equation 21) has the square matrix of coefficients of dimension $k_{1}+k_{2}$ and allows a single solution to be obtained. The method based on the use of (Equation 20) will be called the conditional extremum method with constraints in the form of equations of program reactions or simply the conditional extremum method.

This method allows taking into account $k_{2}-k_{1}$ more of the constraint equations. The linear combinations of forces in the actuators (Equation 13) can be used as these constraints. In this case, expression (Equation 20) will be
(25)$$I^{*}=p_{2}^{T}V_{2}p_{2}+\left({\mathrm{Ch}}_{2}\left(t\right)-\varphi \left(t\right)-p_{10}\right)\lambda +V_{3}\left(p_{20}+{\mathrm{Bh}}_{2}\left(t\right)\right)\lambda_{1},$$
where $V_{3}$ is the matrix of weights with dimension $k_{3}\times k_{2},k_{3}\leq k_{2}-k_{1}$ and $\lambda_{1}$ is the corresponding vector of Lagrange multipliers of dimension $k_{3}$. The linear system, Equation (Equation 21), for the functional (Equation 25) will have the following form
(26)$$\left(\begin{array}{ccc} B_{1} & C^{T} & B_{2}^{T}\\ C & 0 & 0\\ B_{2} & 0 & 0 \end{array}\right)\left(\begin{array}{c} h_{2}\\ \lambda \\ \lambda_{1} \end{array}\right)=\left(\begin{array}{c} b_{21}\\ b_{22}\\ b_{23} \end{array}\right),$$
with the matrix $B_{2}$ as
(27)$$B_{2}=V_{3}B,$$
and the matrix $b_{23}$ as
(28)$$b_{23}=-V_{3}p_{20}.$$

The method based on the use of (Equation 25) will be called the conditional extremum method with constraints in the form of forces in the actuators. If $k_{3}+k_{1}=k_{2}$, system (Equation 26) splits into two independent systems. The column matrix $h_{2}$ is unambiguously determined from the second and third equations of (Equation 26), which are similar to system (Equation 17). Function (Equation 25) is close to the function of the Karush–Kuhn–Tucker (KKT) method, which is well-known in the theory of nonlinear programming [8]. However, in our case, it is used without any additional conditions that are used in the KKT.

The conditional extremum method makes it possible to reduce the optimal control problem to the parametric optimization problem over the state space of relatively small dimensionality, compared with the direct parameterization of the control functions.

## 3. Case Studies

The method considered in Section 2 is implemented in the MBS dynamics software FRUND [20]. The examples below are solved using it.

### 3.1. Inverted Double Pendulum

Consider the flat inverted double pendulum shown in Figure 1. These pendulums are often considered in problems of control synthesis of walking robots [21,22,23,24,25,26,27,28,29,30,31,32]. The limit on the magnitude of the torque in the pendulum support is a condition for the stability of the robot (i.e., avoiding overturning). The problem is to find the law of change of torque at points *B* and *C* with an arbitrary law of motion of point *A*. Simple usage of the constraint equations in one or two directions at point *A* leads either to the fixation of the pendulum in its original position or to the fall of the pendulum if one connection is specified in the horizontal direction.

Various options for finding control torques can be considered within the framework of the proposed method. The simplest case is specifying the constraints at point *A* along the vertical coordinate $Z_{A}=0$. The control torque will be found only at point *B*. The parameters of the dimension that was introduced in Section 2 are n=6, k=6, $k_{1}=1$, $k_{2}=1$; the control torque $M_{B}$ is found from Equation (Equation 13). The motion picture of the pendulum is presented in Figure 1b. The plot of the torque change $M_{B}$ is presented in Figure 2. During the calculations, it was assumed that the control torque smoothly reaches the program’s preset value in 0.05 seconds. The sharp increase in the control torque at the end of the movement is explained by the approach to the singularity position—aligning the pendulum links along the same line (see Figure 1b). Therefore, in this simplest case, the problem of determining the control torque is solved unambiguously.

### 3.2. Spatial Model of Android

Let us consider the spatial motion of a mechanical system on the example of an android robot. The calculation scheme of such a robot is presented in Figure 3. The system parameters are n=150, k=144. The number of actuators in the android structure is 21. The calculation scheme contains two masses with large values of inertial parameters to determine 12 reactions in the contacts of the android’s feet with the supporting surface. This method allows for solving some special cases of systems with redundant constraints. In this case, the redundant reactions are six reactions in the feet. Calculations were made for the variant of 8 control drives and restrictions on 6 reactions—$k_{1}=6$, $k_{2}=8$. Control drives are rotation drives working around the transverse axis of the robot in the hinges of its shoulders, hips, knees, and feet—two for each type of hinge. We modeled the displacement of the center of mass of the robot back by 2 cm in 2 s. Vertical reactions and reaction torques were considered unchanged relative to the transverse axis. Horizontal reactions in the feet were calculated from the horizontal inertia forces caused by the movement of the center of mass. As the torque of the reaction was set to be unchanged while the static torque of this reaction increased due because of the movement of the center of mass, this change was compensated for by the movement of the robot’s sections, in particular, by the rotation of its arms (see Figure 4). This movement corresponds to the natural reaction of a human when trying to maintain balance without being able to move their legs. The results shown in Figure 4 are obtained by resolving the ambiguity using the conditional extremum method—Equation (Equation 21). The squares of angular velocities in the corresponding hinges were taken as weights. During calculations with the same unit weights, the torque compensation occurred solely because of the movement of the body.

The considered example of controlling an android robot as a multibody mechanical system allows the conclusion that the method is sufficiently versatile and applicable to a wide class of mechanisms to be made, for example, for parallel mechanisms [33,34,35,36,37].

### 3.3. Optimal Control Problems in the Example of Car Vibrations

Optimization criteria such as (Equation 2) are widely used in practical applications of mathematics and mechanics [38,39,40]. Consider the classical problem of controlling a vibration-isolating system using the car suspension example. Minimization criterion (Equation 2) can be presented in the proposed approach as follows
(29)$$J=\int_{0}^{T}R^{*}\left(p_{1}\right)dt.$$

As $R^{*}$ is a function of some program values of Lagrange multipliers $p_{1}$, which are assigned to desirable functions φ(t), the sum of the components of vector φ(t) can be used as $R^{*}$ function. The minimum of functional (Equation 29) is achieved, for example, by functions φ(t) being equal to zero. The greedy control criterion (Equation 3), in this case, will take the following form
(30)$$I=\sum_{i=1}^{i=k_{1}}\varphi_{i}\left(t\right).$$

The controls for criterion (Equation 30) can be found by using (Equation 10)–(Equation 25). Let us emphasize that the control obtained from criterion (Equation 30) is greedy control. However, it is the optimal control as well because it provides the minimum of criterion (Equation 29).

Consider the problem of controlling a car’s suspension to reduce its vibrations from the impact of the road’s micro profile. The existing problem statements can be found in [41,42,43,44]. Figure 5 shows the calculation scheme of the mathematical model of the car, which makes simulating its movement along the road irregularities possible. We simulated the movement of the car through a triangular irregularity. Vertical accelerations in the front of the car are presented in Figure 6. In the controlled version, two connections are set—zero vertical movements at two symmetrical points, *A* and *B*, in the front of the car body. The M1 and M2 torques in the two front suspension arms are used as actuators. The dimension parameters are $n=96,k=85,k_{1}=2,k_{2}=2$. Using (Equation 12), we determined the control torques in the suspension levers, ensuring the movement of points on the frame with zero reactions. Frame accelerations at the considered points are close to zero (see Figure 6). This problem can be considered an example of solving an optimal control problem with the optimization criterion in the form of zero displacements of the selected points of a mechanical system.

## 4. Conclusions

The proposed method of calculating control can be considered a universal theoretical method for solving a wide range of problems related to controlled system dynamics, including the problems of controlling robot manipulators, anthropomorphic and zoomorphic robots, vibration damping problems, etc. The important feature of this method is that it is based on numerical models of machine dynamics, which are widely used in existing computer simulation programs for the dynamics of mechanical systems. The method has no fundamental limitations on the dimensionality of modeled systems and types of nonlinearities.

The evaluation of the proposed method on the described use cases and other test examples proved that computational efficiency has increased for all problems described by DAE (differential-algebraic equations). It was achieved for DAE with a wide range of state dimensions—from 12 to 180 ($k_{1}+k_{2}$) and control dimensions from 1 to 8. The dimensionality of the parameter space is independent of the state dimension and defined only by the number of controls.

The proposed method is a universal theoretical method for the optimal control problem of the systems meeting the following requirements:the system is described by DAE (Equation 2), which has numerical solutions; constraint Equation (Equation 1) is a function of coordinates (holonomic constraints in mechanics);the integral object function contains only Lagrange multipliers (Equation 29);matrix A is not singular;the linear system in Equation (Equation 11) is joint, i.e., it has at least one solution.

The important feature of this method is that it can be considered a kind of machine learning, in particular, reinforcement learning, as a variational formulation of the control problem. The formulation of the proposed method in the form of functionals (Equation 20) and (Equation 25) corresponds to the so-called “greedy” control [14] in reinforcement learning methods and, at the same time, is the optimal control with the appropriate formulation of integral optimality criteria. From this point of view, the considered method can significantly reduce the dimensionality of the parameter space, and consequently, increase the computational efficiency of machine learning methods.

The purpose of the presented method is to provide the reference optimal trajectories and controls in the case of the agent having complete knowledge of the environment. The stability problem, controller optimization, and uncertainty model fall beyond the borders of this study. For the tasks of robot control, the standard methods of achieving robustness can be used [45].

This work presents the fundamental theoretical provisions of the method and does not address such issues as control stability and control in systems with singular matrices. These issues are the subject of further research.

## Figures and Tables

**Figure 1 sensors-22-08920-f001:**
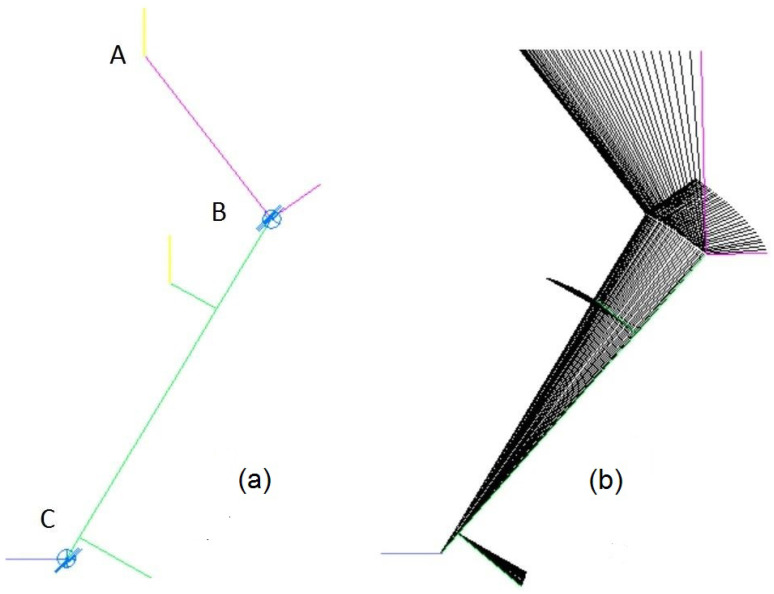
Calculation scheme (**a**) and motion picture (**b**) of the fall of the inverted double pendulum with the condition of horizontal movement of a given point A. A is the point for which reference motion is defined. B and C are pendulum links.

**Figure 2 sensors-22-08920-f002:**
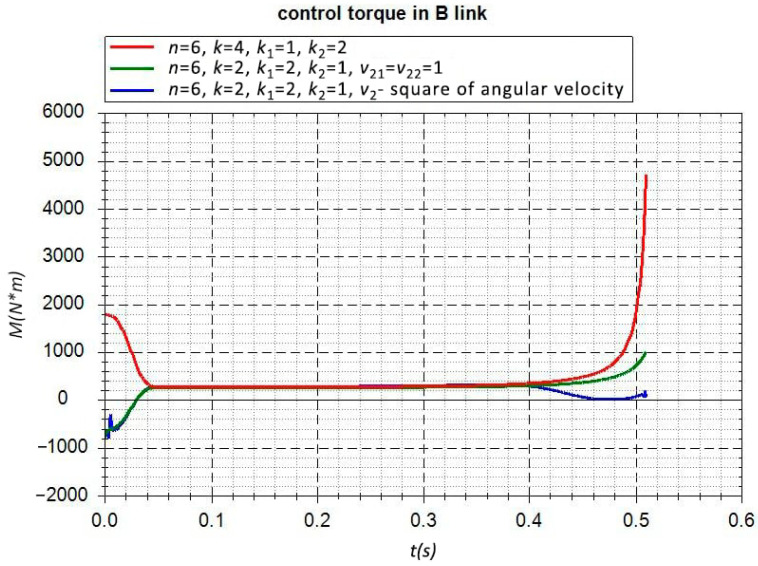
The control torque in link *B* when the horizontal movement of point *A* is free with various control options.

**Figure 3 sensors-22-08920-f003:**
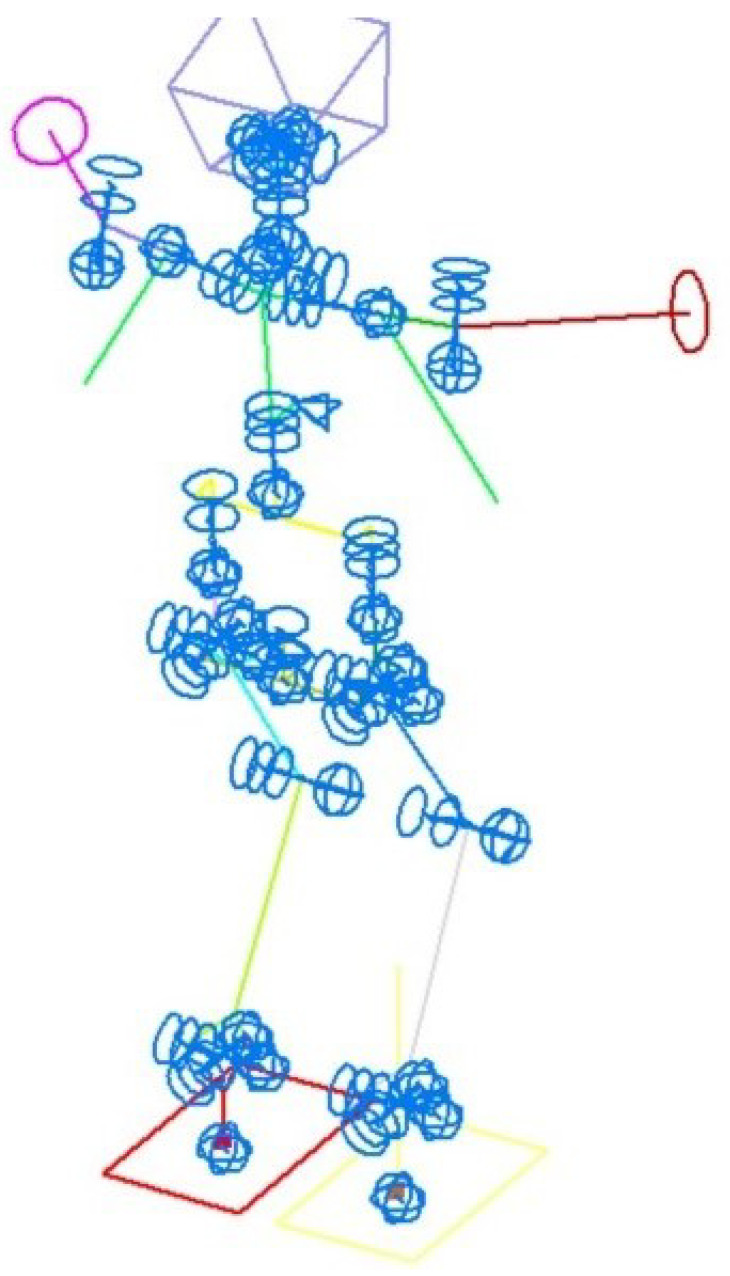
The calculation scheme of an android robot. Blue markers are the links; other colors mark the different bodies of the system.

**Figure 4 sensors-22-08920-f004:**
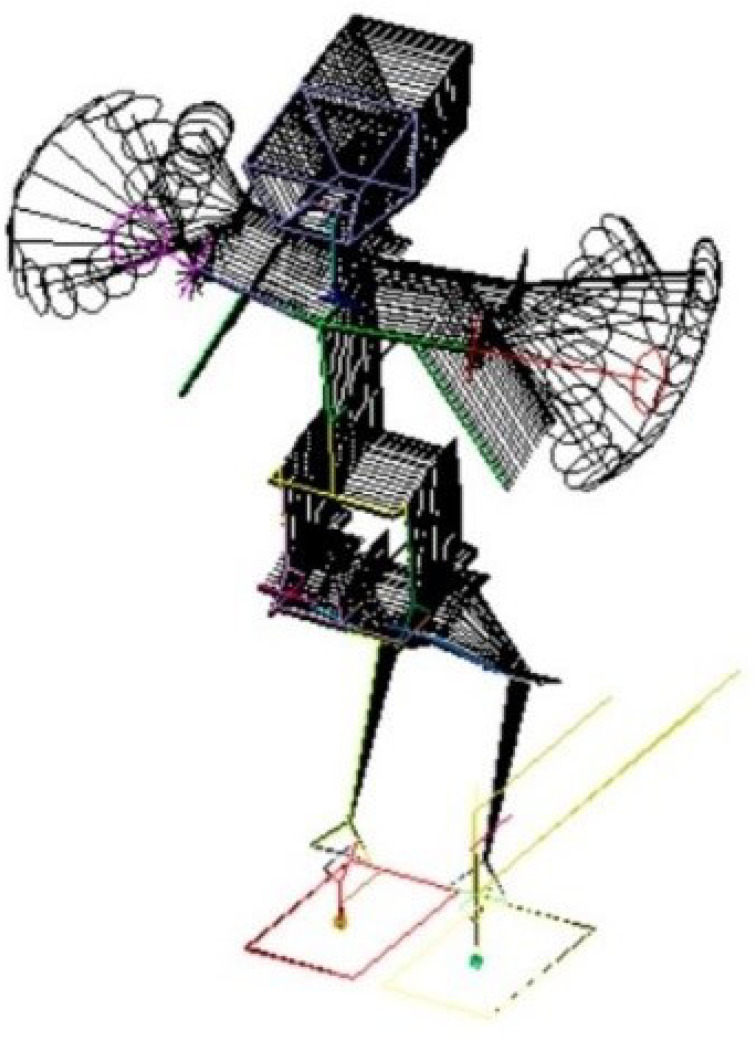
The movement picture of the android’s movement when the center of mass is shifted backward while maintaining the magnitude of the reaction torque in the support relative to the transverse axis.

**Figure 5 sensors-22-08920-f005:**
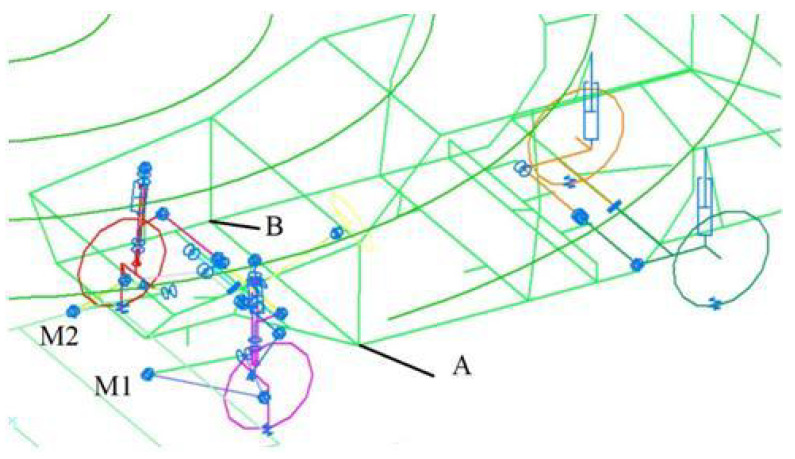
The calculation scheme of a car. A and B is the points whose reference motion is defined. M1 and M2 are the links where the control torques are applied.

**Figure 6 sensors-22-08920-f006:**
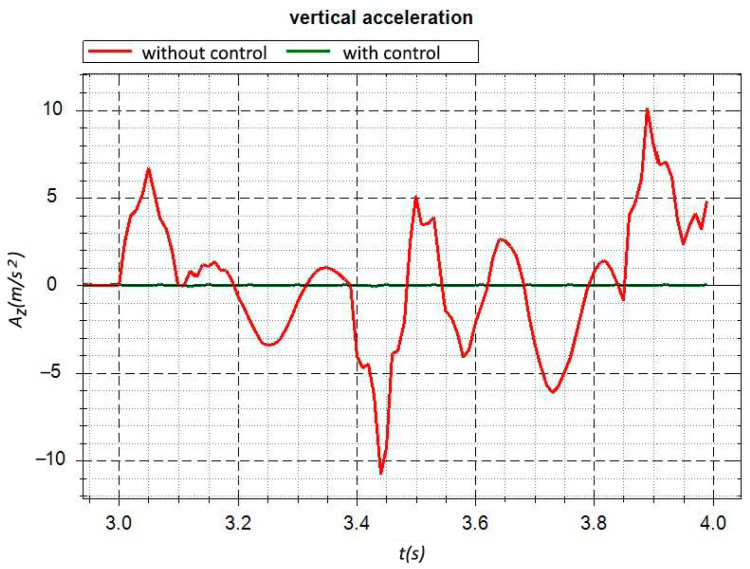
Vertical acceleration in the car’s front part.

## Data Availability

The source code of the developed program FRUND can be accessed freely at svn://dump.vstu.ru/frund.

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
