# Peer review of "Optimal Greedy Control in Reinforcement Learning"

_sensors, 2022, doi:10.3390/s22228920_

Round 1
Reviewer 1 Report
The work presented is related to controlled dynamics and is supported by case studies.
1. It is stated that, reduction of dimensionality parameter space has increased computational efficiency is this relates only with respect to the case studies or in most of the cases.
2. Need to justify further properly that it is a universal theoretical model for solving wide range of control dynamics.
Author Response
- It is stated that, reduction of dimensionality parameter space has increased computational efficiency is this relates only with respect to the case studies or in most of the cases.
We added the following paragraph to Conclusion:
The evaluation of the proposed method on the described use cases and other test examples proved, that computational efficiency has increased for all problems, described by DAE (differential algebraic equations). It was achieved for DAE with a wide range of state dimensions -- from 12 to 180 ($k_1+k_2$) and control dimensions from 1 to 8. The dimensionality of the parameter space is independent from the state dimension and defined only by the number of controls.
2. Need to justify further properly that it is a universal theoretical model for solving wide range of control dynamics.
We added the following paragraph to Conclusion
The proposed method is a universal theoretical method for optimal control problem of the systems meeting the following requirements:
• the system is described by DAE (2) that have numerical solutions; the constraint equations (1) are functions only of coordinates (holonomic constraints in mechanics);
• integral object function contains only Lagrange multipliers (29);
• the matrix A is not singular; • the system of linear equations (11) is joint. i.e., it has at least one solution.
Reviewer 2 Report
In this paper, the application of variational method in control design of robotics has been proposed. The topic is interesting and the simulation results are abundant. However, My major concerns are listed as follows:
1. The dynamic models of inverted pendulum, car, android robot should be given as well as the application of presented theory in chapter 2 to obtain the proposed controller for car, android robot should be added more;
2. The variational method is focused on to present but the Reinforcement Learning technique has not been presented sufficiently. Moreover, the comparisons with previous references considering application of RL in Robotics need to be discussed, such as https://ieeexplore.ieee.org/abstract/document/8396836, https://www.sciencedirect.com/science/article/pii/S0019057822001495.
3. Please discuss the posibility of the proposed method in the case of Uncertain Model.
Author Response
- The dynamic models of inverted pendulum, car, android robot should be given as well as the application of presented theory in chapter 2 to obtain the proposed controller for car, android robot should be added more;
The dynamic test models have numerical form (see lines 50-58, [13-14]) and is too big to be given in the article text. The simplest test model of inverted pendulum consists of 12 nonlinear equations.
We can put the inverted pendulum model in the appendix if necessary.
2. The variational method is focused on to present but the Reinforcement Learning technique has not been presented sufficiently. Moreover, the comparisons with previous references considering application of RL in Robotics need to be discussed, such as https://ieeexplore.ieee.org/abstract/document/8396836, https://www.sciencedirect.com/science/article/pii/S0019057822001495.
The proposed method solves the optimal control problem, which is a type of RL problem with complete knowledge according to [14].
The works in the references https://ieeexplore.ieee.org/abstract/document/8396836 and https://www.sciencedirect.com/science/article/pii/S0019057822001495 study the stability problem of perturbed control motion for known referenced motion. Our study is aimed at finding the referenced motion. We added this discussion to the article (see lines 20-22).
3. Please discuss the posibility of the proposed method in the case of Uncertain Model.
We added the following text to the Conclusion
The purpose of the presented method is to provide the reference optimal trajectories and controls in case of the agent having complete knowledge of the environment. The stability problem, controllers optimization, and uncertainty model fall beyond the borders of this study. For the tasks of robot control, the standard methods of achieving robustness can be used [46].
Reviewer 3 Report
There are no MAIN questions opened because the proposed work is, in my opinion, complete and solid. For a better understanding of the material, I would suggest the Authors address these MINOR points:
A. Fig. 3: the figure is interesting but difficult to understand. The Authors are invited to add a better explication in the text and/or in the caption.
B. Fig. 6: the small little bumps in Fig. 6 are not readable. The Authors are invited to add a sort of Fig. 6b in which the green curve ALONE can be appreciated in its vertical scale.
Author Response
A. Fig. 3: the figure is interesting but difficult to understand. The Authors are invited to add a better explication in the text and/or in the caption.
We enhanced the caption of Figure 3.
B. Fig. 6: the small little bumps in Fig. 6 are not readable. The Authors are invited to add a sort of Fig. 6b in which the green curve ALONE can be appreciated in its vertical scale.
We increased the size of Figure 6 to make the plots more readable.